



# Measurements of aeolian sediment transport in the vicinity of coastal dunes on Spiekeroog Island, Germany, and extrapolation to annual transport volume

Malte Kumlehn[1], Oliver Lojek[1], Viktoria Kosmalla[1], Björn Mehrtens[1], Lukas Ahrenbeck[1], David Schürenkamp[1], and Nils Goseberg[1,2]

[1]Division of Hydromechanics and Coastal Engineering, Leichtweiß-Institute for Hydraulic Engineering and Water Resources, Technische Universität Braunschweig, Beethovenstraße 51a, 38106 Braunschweig
[2]Coastal Research Center, Joint Central Institution of the Leibniz Universität Hannover and the Technische Universität Braunschweig, Hannover, Germany

**Correspondence:** Malte Kumlehn malte.kumlehn@tu-braunschweig.de

**Abstract.**

This work presents a field study measuring aeolian sediment transport and wind profiles across a dune, and provides an approximation of the annual volume of aeolian transport into the dune systems of the East Frisian island of Spiekeroog, using commonly available meteorological data. Aeolian sediment traps were positioned along a transect aligned with the general wind direction during the measurement, starting on the open beach and ending on the back site of a selected dune. Wind profiles were temporarily measured next to four aeolian traps. Using a recent version of an aeolian sediment transport model, the total annual aeolian sediment transport into the dune systems is approximated. Input variables for this model are the time series of shear velocity and surface moisture. These are derived from the wind velocity measured at a height of 10 m and the amount of precipitation in combination with potential evaporation calculated using radiation intensity. The results are compared to volume changes of the beach and dune systems, which are derived from geospatial data. Data from the field study shows, that sediment transport occurs even behind twenty metres of vegetation on top of a dune. Further, the study indicates that the impact of precipitation on aeolian transport reduction can be lower within vegetated areas on a dune compared to the open beach. The approximation of the total annual aeolian transported sediment surpasses the actual volume changes of the dune systems as expected, however this difference varies depending on the compared beach section almost by a factor of five.

## 1 Introduction

Aeolian sediment transport is a key driver for the initiation and growth of coastal dunes (Bagnold, 1974). Initially, aeolian sediment transport processes were described using straightforward approaches based around an equilibrium system, where the transport rate is approximated as a function of wind velocity, also called shear velocity, density of transported and transporting mediums (air/sediment) and sediment grain size (Ralph Alger Bagnold, 1937; Horikawa and Shen, 1960; Kadib, 1964; White,





20 1979). Field measurements showed that this basic model performed moderately well, within orders of magnitude regarding estimated transport volumes for select sites (Horikawa and Shen, 1960; Arens, 1996; Sherman et al., 1998; Kroon and Hoekstra, 1990). Consequently, prediction equations describing aeolian transport were further refined to account for ambient moisture (Belly, 1964; Mckenna Neuman, 2003), beach slope (White and Tsoar, 1998) and pebble lag (Nickling and McKenna Neuman, 1995; Davidson-Arnott et al., 1997). Notwithstanding the advancements, calculations for longer time periods still showed sig-
25 nificant deviations from observed volumes (Davidson-Arnott et al., 2012). Direct transport measurements on various coastlines revealed, that the transport rate increases from zero at the top of the swash zone to a maximum downwind (van der Waals and Rowlinson, 1988; Davidson-Arnott and Law, 1996). This maximum corresponds to a theoretical upper limit of transport capacity, when sufficient material supply is available and the fetch length is long enough to transfer all the vertical momentum input from the wind velocity into the initiation of saltation of sediment grains on the beach (Nickling and Davidson-Arnott,
30 1990). This increase in transport rate is termed "fetch erosion effect" (Gillette et al., 1980, 1996) or "fetch effect". Surface moisture is a major factor influencing the fetch effect (Belly, 1964; Johnson, 1964). A repercussion of a moist surface can be the formation of a solid crust, potentially due to organic processes or salt precipitation, which in turn hinders the aeolian sediment transport even though the surface of the beach dried up (Johnson, 1964; Lancaster and Nickling, 1994; Gillette et al., 1980; Walker, 2020).

35 Calculating potential aeolian sediment transport uses wind velocity data and is typically based on the law-of-the-wall. Deriving the slope of the time averaged wind velocity profile and using the von Karman constant (von Karman, 1930) allows for estimating the essential shear velocity (Bagnold, 1974; Horikawa and Shen, 1960; van Rijn and Strypsteen, 2020) for transport calculations. Direct measurements of vertical wind velocity profile acquire the overall bed shear velocity, as it automatically includes effects of roughness elements found on a beach as well as morphological bed features such as ripples, beach berms (van
40 Rijn and Strypsteen, 2020). However, defining the shear velocity correctly is key, as it exhibits a major influence on the overall transport volume (Sherman, 2020). Recent literature shows, that considering detailed information regarding the prevalent grain size distribution greatly improves projected transport volumes (van Rijn and Strypsteen, 2020). Sherman (2020) has compiled an extensive overview over the last decades of research and development pertaining to equipment, methods and insights on aeolian transport research and has synthesized six main research theories to be investigated to further improve the scientific
45 basis in the field:

  1. **Transport rate estimates** generally are based on models derived from wind tunnel experiments, which attempt to simulate ideal conditions with constant wind fields, unlimited sediment supply and saturated transport conditions. However, wall effects and wind tunnel facilities are too short to reach saturated transport conditions impinge these efforts (Hong et al., 2018). Furthermore, field studies show that large turbulent flow patterns cause substantial transport variability
50   (Baas and Sherman, 2005; Bauer and Davidson-Arnott, 2014). Until now, a common methodological approach to number, spatial distance and duration of measurements in the field is lacking to accurately map wind parallel and spanwise variability; no theory exists to predict this variability for field conditions and no methodological protocol exists for representing transport measurements.





2. **Grain size characterization** is important for grain-grain related interactions, since size and shape determine hiding functions and armour layer development (Nickling, 1988), in turn influencing critical shear velocities. Furthermore, the grain size distribution curve is method-dependent, as a weight-frequency distribution based on sieving results in a distorted $d_{50}$ is directly impacting the transport rate estimates (Konert and Vandenberghe, 1997).

3. **Density characterization** of fluids and solids involved within the transport equations is important, since environmental conditions vary in the field, whereas lab derived transport equations represent ideal and invariable conditions. In that regard, bulk densities of solids vary depending on the available material. While there is a lack of extensive research, it is evident that the varying density of the transport medium significantly impacts the shear velocity required for transport initiation (Mckenna Neuman, 2003). This in turn will affect the annual aeolian transport, due to the seasonal temperature differences and the resulting changes in air density (Francis and Peters, 1980).

4. **Shear velocity** has to be usually estimated for aeolian transport calculations. General disagreement on the wind-sand-relation resulted in models following Bagnold to use the shear velocity $u^*$ to the power of 3, while other models use a power of 2, in turn amplifying small errors in $u^*$ affecting transport volumes. Furthermore, the value of the van Karman constant $\kappa$ has been shown to decrease with increased sand transport (Strypsteen and Rauwoens, 2023) introducing another error source. The roughness length $z_0$ is derived using the approach by (Nikuradse, 1931), however saltation has been shown to further increase $z_0$ with increased $u^*$ reducing estimate reliability.

5. **Initiation of sand grain motion** in literature is usually defined with various views, i.e., on "single grains" as well as for "patches", "constant saltation", and eventually up to "a complete sediment bed in motion" (Swann et al., 2020); these separate views on what specifically moves exhibiting a large variance and hence may depict a large error source. Furthermore, moisture has been shown to have a major impact on the transport. Notwithstanding its impact on overall transport, it is often omitted or empirically calibrated (Nickling, 1988; van Rijn and Strypsteen, 2020).

6. **Granular electrification** is an aspect largely omitted in aeolian transport calculations in literature, as little research is available on the subject (Sherman, 2020).

Continuing onwards from the theoretical research gaps briefly summarized from Sherman (2020) and other relevant literature, research reports that prevalent wind direction and speed in combination with sediment availability and pioneering vegetation additionally play a key role in the creation of incipient dunes (Adriani and Terwindt, 1974; Buckley, 1996). Another aspect exhibiting a large influence is the available fetch length between the tidal high water line and the dune foot, usually in the form of a beach. The distance along which wind can pick up dry sediment grains limits the degree of saturation of a theoretical transport capacity due to wind speed (Bauer and Davidson-Arnott, 2003).

Empirical transport models predicting the aeolian transport have been derived and applied to estimate dune growth at different coastlines with varying degrees of accuracy (Davidson-Arnott and Law, 1996; Hesp et al., 2005; Homberger et al., 2024; van Rijn and Strypsteen, 2020; Kroon and Hoekstra, 1990; Shao, 2009; Strypsteen, December 2019). Furthermore, details on grain-related shear velocity and bed roughness have been shown to exhibit a major impact on the projection of wind-driven





beach dune growth and may result in a substantial improvement of sedimentation volumes (Strypsteen and Rauwoens, 2023; Strypsteen et al., 2024b)

Apart from the physical processes and environmental conditions governing aeolian transport, the surface topography within
coastal dunes often is at least partially covered with typical dune vegetation such as marram grass (Hesp et al., 2005; Nield and Baas, 2008; Goldino et al., 2024; Strypsteen et al., 2024b; Biel et al., 2019) or other vegetation such as shrubs or trees (Provoost et al., 2011). The influence of the presence of vegetation on dune surfaces to the aeolian sediment transport has been the subject of various studies (White, 1979; Biel et al., 2019; Davidson-Arnott et al., 2012; Goldino et al., 2024; Homberger et al., 2024; Rotnicka et al., 2023; Strypsteen et al., 2024a). In summary, it was found that depending on the incident wind
angle and velocity, sediment transport occurs within the dune vegetation canopy or skimms above it at a certain velocity (ebda). Downwind of the dune system, transport dissipated quickly. Three established modes of transport associated to wind flow have been defined, which have been associated to lateral upwind canopy coverage percentages and pertain to (1) isolated roughness flow with up to $< 16\%$, (2) wake interference flow with coverage degrees from $16 - 40\%$ and (3) skimming flow with coverage rates $> 40\%$. A fourth mode was suggested by Hesp et al. (2019) as canopy flow taking place inside the vegetation for areas
exhibiting coverage degrees $20 - 40\%$.

The majority of studies investigate the aeolian transport process as a whole or in parts, employing focused laboratory studies (Eichmanns and Schüttrumpf, 2022; Han et al., 2011; Li and Mckenna Neuman, 2014) as well as field studies deploying profile setups of sand traps and anemometers (Arens, 1996; Baas and Sherman, 2005; Bauer et al., 2009; Buckley, 1996; Davidson-Arnott et al., 1997, 2012; Eichmanns and Schüttrumpf, 2020; Eichmanns et al., 2021; Goldino et al., 2024; Jackson
and Nordstrom, 1998; Johnson, 1964; Kroon and Hoekstra, 1990; L. C. Van Rijn, 2018; Nickling and Davidson-Arnott, 1990; Strypsteen, December 2019; Strypsteen et al., 2024b; White and Tsoar, 1998). Furthermore, a couple of studies combined episodic transport measurements (J. Alcántara-Carrió and I. Alonso, 2002; Strypsteen and Rauwoens, 2023) with regional long-term wind data to analyse morphodynamic dune system responses and quantify transport volumes. Empirical modelling approaches are also aiming at shedding light on transport mechanics and controlling environmental variables (Nield and Baas,
2008; Mir-Gual et al., 2023). Some studies have drawn on historic remote sensing data and correlation with matching wind data (Doyle et al., 2019; Galiforni Silva et al., 2019). Jackson and Nordstrom (2011) reviewed dune management methods and highlights the importance of deepening knowledge of quantifying aeolian transport and impacts man-made structures and occupation through vegetation trampling have on it to mitigate and preserve sandy coastlines. Walker et al. (2017) on the other hand, showcase a decadal research program on dune morphodynamics, where they emphasize that coastal dune
systems are situated at the land-sea interface and morphodynamics, therefore are highly scale dependent from local single dune measurements to regional landscape scales of systems and finally up to supra-regional landform scales of coastlines. They stress that the majority of transport calculation approaches used to date, such as the prominent resultant drift potential (Fryberger et al., 1979) are inadequate to capture the spatio-temporal vicissitude of beach-dune transport processes accurately. In similar manner Farrell et al. (2023) compiled a review on contemporary dune and aeolian transport research; this author carved out
some more potential research gaps, i.e., (1) vegetation related roughness effects on aeolian transport and (2) upscaling of



episodic local measurements towards larger and longer spatio-temporal scales. While there remains a number of aspects related to coastal dunes, and their evolutionary processes, unaddressed in the pertinent literature, this study aims to focus on the following specific objectives:

1. To conduct and to test correlation of episodic, local aeolian transport and wind profile measurements for dune faces with
and without vegetation

2. To assess the relation between ground truthing with regional long term weather observations and calculate volume fluxes

3. Test transport rates for sensitivity regarding precipitation, sunshine duration and wind direction

4. To compare dune and beach volume changes derived from federal survey data

5. To upscale aeolian transport rate estimates and to project volume changes for a tidal barrier island

The manuscript is organized as follows, chapter 2 gives an overview of the focus area as well as used data and methods, chapter 3 presents the findings with chapter 4 discussing and interpreting them. Chapter 5 draws a conclusion and gives an outlook regarding future work.

## 2   Materials and Methods

### 2.1   Focus Area

Field measurements for this study were performed on the East Frisian island Spiekeroog, Germany, at the end of May 2023. Spiekeroog is one of seven tidal barrier islands in a chain of islands oriented along a west-east trajectory approximately two km in front of the mainland, see figure 1b. Spiekeroog, like most of the East Frisian barrier islands, features an armored, hard-protected western tip and an elongated sand spit to the east. Spiekeroog spans 10 km in east-west direction and 2 km north-south. With their coastal parallel orientation, the East Frisian islands serve as quasi wave breakers and therefore contribute to
the protection of the mainland (NLWKN, 2010). Long-shore sediment transport from west to east has led to sandy beaches with adjoining dune systems stretching over multiple kilometres on the northern side of the islands (Herrling and Winter, 2018). Whereas, on the western tip of Spiekeroog Island, a deficiency in sediment accumulation exists. Groins and a masonry protection wall are stabilizing the base of the dunes in this specific region (Hanisch, 1981). Conversely, across the remainder of the island, the inherent dune system serves as the sole protective barrier along its northern, seaward-facing shoreline. Wind
directions change throughout the year, with predominantly southwestern winds in the winter, northwestern winds from spring to summer and southern winds in autumn, as shown in figure A1 for the year 2022. Furthermore, the highest wind velocities are present in the winter and spring. The site for the field measurements encompasses a freestanding cusp-dune situated at the central portion of the northern coast, which constitutes a part of the primary dune row in that vicinity. The dune toe is currently situated at around 1.5 m above Mean High Water (MHW) (figure 1e), implying that dune erosion can only occur during storm
surges. Twenty-nine storm surges occurred over the span of fourteen years between 2009 and 2023, with fourteen classified as





severe storm surges, reaching an inundation $1.5\ m$ over MHW (BSH, 2024). Tidal range spans 2.71 m, regularly inundating extensive areas of the beach located below mean tidal high water. The island features established and partially stabilized gray dunes along the south-west and west parts, ranging in elevation from $7.35 - 9.45$m with a maximum of $11.24$m for the south-west and from $10.72 - 16.50$m on the western shoreline with a local maximum of $20.73$m. Towards the eastern part of the sand

spit, the dunes along the northern beach are relatively young white dunes, which developed with the expansion of the island to the east since the 1980s. The elevation ranges from $3.5$ to $12.0$m with a mean of $5.5$m.



**Figure 1.** (a) Area of the German Bight with EMODNET Bathymetry 2020 (EMODnet Bathymetry Consortium), (b) Spiekeroog digital orthophoto with beach sections and the approximate locations of the field measurements (LGLN, 2024), (c) Digital Elevation Model (DEM) of focus area based on composite of public data (GDWS, 2021a, b), (d) location of the cross-shore bathymetric-topographic profile extracted from DEM, (e) elevation along a cross-shore profile at the study site.



## 2.2 Measurements of wind velocities and aeolian sediment transport

Wind speed and direction are measured at meteorological stations around the globe, following World Meteorological Organization (WMO) standards (WMO, 2014). For this study, such WMO complying data has been obtained from the German Weather
Service (*Ger.: Deutscher Wetter Dienst*) (DWD) using acoustic anemometers (DWD, 2024). By measuring wind velocity at least at three different heights near the surface, we obtain horizontal wind velocity profiles from the wind measurements data (Bauer et al., 1992). Multiple studies used mobile meteorological stations with anemometers installed in a vertical configuration, at times supplemented by wind vanes measuring the direction (e.g. Hesp et al. (2005); Strypsteen (December 2019); Eichmanns and Schüttrumpf (2020)). During this field study, a wind tower consisting of five cup anemometers from PCE In-
struments (type PCE-FST-200-201) and a wind vane (type PCE-FST-200-202) has been used. The anemometer can measure from $0.5\,to\,50\,m\,s^{-1}$ with an accuracy of $\pm 0.5\ m\ s^{-1}$ up to 5 m s$^{-1}$ and 3% above it, whereas the vane aligns itself above 0.8 m s$^{-1}$. While the anemometers were positioned at different heights from $0.2\,to\,4$ m (see. figure A2a), the wind vane was positioned halfway up at 2 m. Data acquisition was undertaken at a sampling rate of 8 Hz during the measurements. Each sensor is connected to the logger via a cable, which is not shown in the figure. For anchoring, the lowest 48 mm are inserted
into the ground, and three steel cables are tensioned at a height of three meters (see. fig. A2b).

Aeolian transport has been measured with a variety of sensors, ranging from fine nets and rigid containers to catch sand midair (Sherman et al., 2014; Wilson and Cooke, 1980; Basaran et al., 2011) to non-intrusive detection using optical detection (Davidson-Arnott et al., 2012) or more established saltiphones amplifying detected sounds of sand grains jumping along the beach (Eichmanns and Schüttrumpf, 2020). Aeolian sediment transport was measured using aeolian sediment traps consisting
of seven Basaran and Erpul Sediment Traps (BESTs) distributed across four separately horizontally rotating sections at heights from 12 to 97 cm (see dimensions in figure A2a). Each section aligns itself in the wind direction with its fin. The lowest tail is angled to avoid a collision with the dune slope during rotation. According to Başaran et al. (2017), the trap has an efficiency of 80 to 100 % depending on the particle sizes and wind velocities. In the lowest section, the BEST traps are closer together, as the highest transport rates are expected near the ground (Bagnold, 1974; Strypsteen, December 2019; Eichmanns and Schüttrumpf,
2022). The main pole extents 50 cm into the ground as an anchor.



**Figure 2.** Photo of the wind tower at position WT1 placed next to the aeolian trap AT1 in front of the freestanding cusp dune.



## 2.3 Field plan and survey program

From the 29th May to the 2nd June 2023, a field campaign was carried out on Spiekeroog. The aeolian traps were deployed along a transect traversing the above-mentioned freestanding cusp-dune in cross-shore direction, aligned with the main wind direction from North to South. A total of 9 individual aeolian traps were distributed along selected key points over the dune

profile. Starting with location AT1 at the open beach in front of the dune. Positioned at the foot was AT2, followed by AT3 closely behind the crest. Two significant other positions are AT6 and AT7, which were placed directly in front and behind a patch of marram grass. The other traps were distributed along the chosen cross-section, mostly for the profile to be conveniently covered with measurement equipment. More specifically, the elevation breakpoints and gradient changes have been picked to enhance our understanding about possible sediment transport processes along these characteristic locations across the profile.

These positions can be seen in figure 1 as top view and in figure 3 as side view. The positions of the aeolian traps were maintained between the measurements. In total, six data sampling intervals lasting each half a day were logged.

Furthermore, wind profiles were measured near four aeolian traps using the wind tower to retrieve, among other properties, the parameter of shear velocity. Each measurement lasted 30 min and was repeated four times during the field campaign.

The resulting shear velocities and zero roughness length are compiled in table 1 shows

An overview of the times is compiled in table A1 and the positions are shown in figure 3.





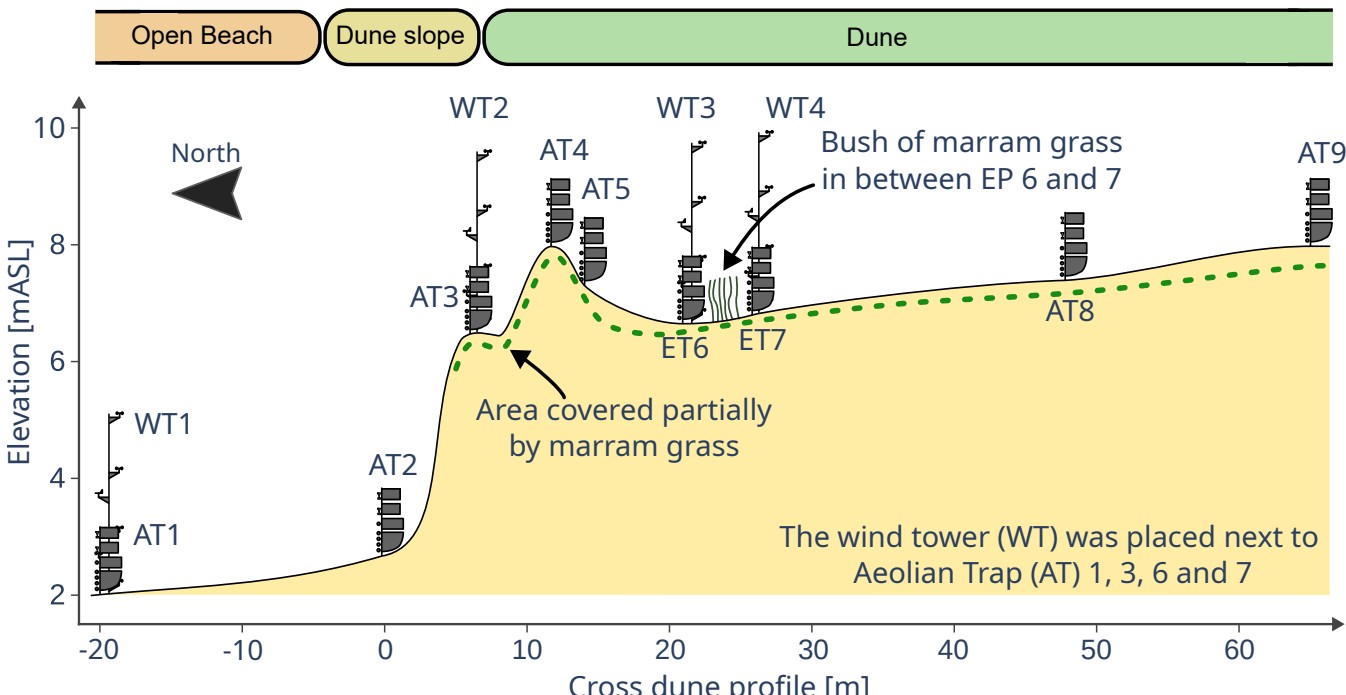

**Figure 3.** Cross-section of the single dune with the positions of the aeolian traps (AT) and the wind tower (WT) relative to the dune foot. The height of the dune is 1:3 exaggerated to the width.

**Table 1.** Mean shear velocity and zero roughness length of the field measurements

| Measurement | Date & Time | $\bar{u}_*$ | $\bar{z}_0$ |
|---|---|---|---|
| | | m/s | mm |
| W20230531.1 | 2023-05-31 10:53 | - | - |
| W20230531.2 | 2023-05-31 18:11 | 0.319 | 1.320 |
| W20230601.1 | 2023-06-01 09:09 | 0.435 | 2.974 |
| W20230601.2 | 2023-06-01 16:47 | 0.382 | 1.596 |
| averaged | | 0.379 | 1.963 |

## 2.4 Methodology for data analysis

Most empirical aeolian transport models use the shear velocity as the main forcing parameter (Strypsteen, December 2019). With at least three measurements at different elevations, it is possible to determine this shear velocity accurately (Bauer et al., 1992). It can be derived from boundary layer theory by fitting the law-of-the-wall equation to wind profile measurements:



$$u_z = \frac{u_*}{\kappa} * ln\left(\frac{z}{z_0}\right) \tag{1}$$

where $u_z$ denotes the wind velocity at $z$, depicting the vertical coordinate, measured from the local bottom of the vertical profile, $\kappa$ is the van Karman's constant, assumed as 0.4. The fitting of this equation results in the shear velocity ($u_*$) and zero roughness length ($z_0$).

Vertical distribution of the aeolian transport flux on the open beach can be described by the following empirical equation 205 (Horikawa and Shen, 1960; Williams, 1964):

$$q_z = q_0 * e^{-\beta * z} \tag{2}$$

with $q_z$ ($kg\ m^{-2}\ s^{-1}$) being the transport flux at the height $z$, $q_0$ ($kg\ m^{-2}\ s^{-1}$) the corresponding transport rate at the surface and the decay rate $\beta$ ($m^{-1}$). This sediment transport flux $q_z$ can be derived from the measurements for each BEST trap by dividing the weight of caught sand by the trap's inlet area ($0.012m * 0.02m$) and by the time delta of the measurement. By 210 fitting equation 2 to the measured data points of $q_z$ and integrating the result over the vertical coordinate $z$, a transport rate $Q_S$ per meter width ($kg\ m^{-1}\ s^{-1}$) is obtained. This method has been demonstrated to be applicable for transport on the open beach at AT1 (Williams, 1964). For measurements where the vertical distribution pattern is not known, linear data point interpolation is used, and the area below is calculated as trapezoids. This is required for measurements on top of the dune (AT2 to AT9):

$$Q_S = \begin{cases} \int_0^{\inf} q_z * dz = \dfrac{q_0}{\beta} & \text{open beach} \\ \sum_{i=1}^{7} \dfrac{q_{z,i} + q_{z,i-1}}{2} * (z_i - z_{i-1}) & \text{on the dune} \end{cases} \tag{3}$$

where $q_{z,i(-1)}$ is the measured sediment flux at the corresponding height $z_{i(-1)}$ above the ground.

Empirical equations to predict aeolian transport rates $Q_S$ have been compared in Strypsteen (December 2019) to data from three field campaigns. The following model proposed by L. C. Van Rijn (2018) and further refined in van Rijn and Strypsteen (2020), which is a modified version of the Ralph Alger Bagnold (1937) showed the best overall fit.

$$Q_D = \alpha_B \alpha_D \alpha_{shell} \alpha_{ad} \sqrt{\frac{d_{50}}{d_{50,ref}} \frac{\rho_{air}}{g}} [(u_*)^3 - (u_{*,th})^3] \tag{4}$$

with the coefficient $\alpha_B = 2$ based on van Rijn and Strypsteen (2020) and the variables $\alpha_{shell}$ and $\alpha_{ad}$ as reduction factors if shells are present at the beach and the width of the beach is below a critical fetch length. Further, $d_{50}$ is the medium grain size



of the beach, $d_{50,ref}$ the reference grain size set as $0.25$ mm and $\rho_{air}$ the air density ($1.2$kg m$^3$). The $d_{50}$ is derived from eight sieve curves, shown in figure A3.

This work additionally uses the factor $\alpha_D$; this denotes the cosine of the angle between the wind direction and the normal to
the dune system orientation, after Bauer and Davidson-Arnott (2003). The normal to the dune system is here defined as the normal to the average orientation of each beach section (North: 0°N, West: 330°N and South: 220°N). For cases when the cosine yields negative values, the factor $\alpha_D$ is set to zero. This way, only the transport into the dune system is calculated. The above outlined model eventually yields the transport rate into the adjoining dunes at equilibrium conditions.

Aeolian transport occurs under the condition that a threshold shear velocity or incipient velocity is exceeded. The shear velocity
at threshold conditions is described by the following equation by Bagnold (1974), with the constant $\alpha_{th} = 0.11$ based on work by Shao and Lu (2000); Han et al. (2011); L. C. Van Rijn (2018):

$$u_{*,th} = \alpha_W \alpha_{slope} \alpha_{th} * \sqrt{\left(\frac{\rho_s}{\rho_{air}} - 1\right) * g * d_{50}} \tag{5}$$

In addition to van Rijn and Strypsteen (2020), his original model is amended by these reductions factor $\alpha_W$ and $\alpha_{slope}$, which depict the moisture content and slope of the surface and the $\rho_S$ is the density of the transported solid.

By knowing the necessary input variables for the aeolian transport rate model above, it is possible to retrieve the annual transport volumes for a whole beach. These input variables for the calculation are shear velocity, and the coefficients $\alpha_W$ and $\alpha_D$. To determine the shear velocity, an approximation must be employed, as there are no direct measurements available. Strypsteen (2023) found a strong correlation derived from wind profiles measurements on the beach and nearby weather station. With this in mind, one can assume the validity for the boundary layer theory up to a height of ten meters and calculate the shear
velocity by using the law-of-the-wall. The annual distribution of the moisture content in the surface of the beach is derived by combining the precipitation with the following evaporation for each ten-minute interval. To calculate the moisture of a next interval ($m_{i+1}$), the evaporation of the current interval ($ET_{P,i}$) is subtracted from the sum of the current moisture content ($m_i$) and precipitation ($P_i$). The resulting $m_{i+1}$ must be in-between zero and the maximum field capacity. Any precipitation which occurs after reaching the field capacity will seep into the ground (WMO, 1992). The surface layer will be assumed to as the
upper 2 mm and the field capacity as 15 % WMO (1958), resulting in a maximum of 0.3 mm of stored water.

$$m_{i+1} = P_i + m_i - ET_{P,i} \quad \text{with } 0 \; mm <= m_{i+1} <= 0.3 \; mm \tag{6}$$

The evaporation during a specific interval $ET_{P,i}$ (mm) is derived by calculating the evaporation $ET_P$ ($mm \; d^{-1}$) using the equation by Turc (1961) divided by 144 10 min intervals per day. The input variables for this equation are the global radiation $R_G$ ($J \; cm^{-2} \; d^{-1}$), the temperature $T$ (°C), and the correction factor $C$ for the humidity $U$ (equation 8).



$$ET_{P,i} = ET_P * (24*6)^{-1} = ET_P = 1.1 * 0.0031 * C * (R_G + 209) \frac{T}{T+15} * (24*6)^{-1} \tag{7}$$

$$C = \begin{cases} 1 + \dfrac{50 - U}{70} & U < 50 \ \% \\[2ex] 1 & U \geq 50 \ \% \end{cases} \tag{8}$$

The zero roughness length $z_0$ can be derived from the wind profile measurements on the open beach which were conducted as part of this study; for the sake of simplicity, we hereby assume the zero roughness length to be constant over the year and seasons. The field survey did not result in a considerable amount of shells covering the beach; thus we set $\alpha_{shell}$ to unity. With the low inclination of the beach up to the dune toe, $\alpha_{slope}$ is also set to unity. When comparing the maximum necessary fetch length of around 100m found by Strypsteen et al. (2024b) to the beach widths found on Spiekeroog (see tab. 3), it can be assumed that there are no significant limitation of the aeolian transport due to a short fetch length. Therefore, the coefficient $\alpha_{ad}$ is also set to unity.

Meteorological data from the DWD and Institute for Chemistry and Biology of the Marine Environment (ICBM) is available in 10 minute intervals to retrieve the above-mentioned variables needed for equation 4. With these, the transport rate $Q_D$ can be calculated for each ten-minute time interval over the year. By multiplying this rate by 600 s, the length of the intervals, one receives the transported volume for each interval, with the summation over the year leading to the annual transported volume (equation 9).

$$V_{transport,annual} = \sum_{i=1.Jan\,00:00}^{31.Dec\,23:50} Q_{D,i} * 600 \tag{9}$$

## 3 Results

### 3.1 Analysis of field measurements of horizontal wind velocity, sediment transport and associated environmental parameters

Along the four positions of the wind-tower measurements (WT1 to WT4; positions are shown in figure 1), different wind profiles can be deduced. Figure 4 depicts one minute averaged vertical profiles of horizontal wind velocity along the measurement transect. The exact profiles varied throughout each 30 minute measurement, but the velocities and the resulting profiles shown in the plots are representative for each measurement. The profiles are amended by adding zero velocity at the elevation of the reference bottom surface and, if feasible, a logarithmic curve was fitted through the data points. Wind profile measurements are named by its date and a trailing number for morning (.1) and for evening (.2) measurements.



Throughout the measurement W20230531.1, an overall lower wind velocity is measured. Above 60 cm, the profile is linear, and the measured wind velocities are the same. Correspondingly, it is not possible to approximate the wind profile for WT1 using a logarithmic function. The absence of a logarithmic profile indicates the applied shear stresses on the surface are very low, which impacts the resulting aeolian sediment transport. Despite being located directly at the seaside crest of the dune, WT2 exhibits a wind profile during this measurement, about reminiscent of a log profile. A similar profile is present at WT3, but starting above the vegetation around $0.7\ m$. For WT4, the first measurement is missing due to a recording error.

The subsequent measurements of WT1 are showing a logarithmic wind profile, which is to be expected on the open beach. These measurements are the source for the values shown in table 1, which are then used in the extrapolation of the aeolian transport. By fitting the equation of the boundary layer theory (equation 1) onto the measurements on the open beach, one can retrieve the shear velocity ($u_*$) and zero roughness length ($z_0$). Table 1 gives an overview of the resulting variables for the three valid measurements. At position WT2, ca. 20 cm behind the edge of the front crest, compression of the streamlines on the dune slope was measured, resulting in the similar high wind velocity for all anemometers except for the lowest. This one shaded by the detachment of the streamlines at the crest. The last two positions, WT3 and WT4, show the wind profile directly before and after a bush of marram grass (see figure 3). The absence of vegetation in front of position WT3 results in a wind profile closely resembling a logarithmic curve.

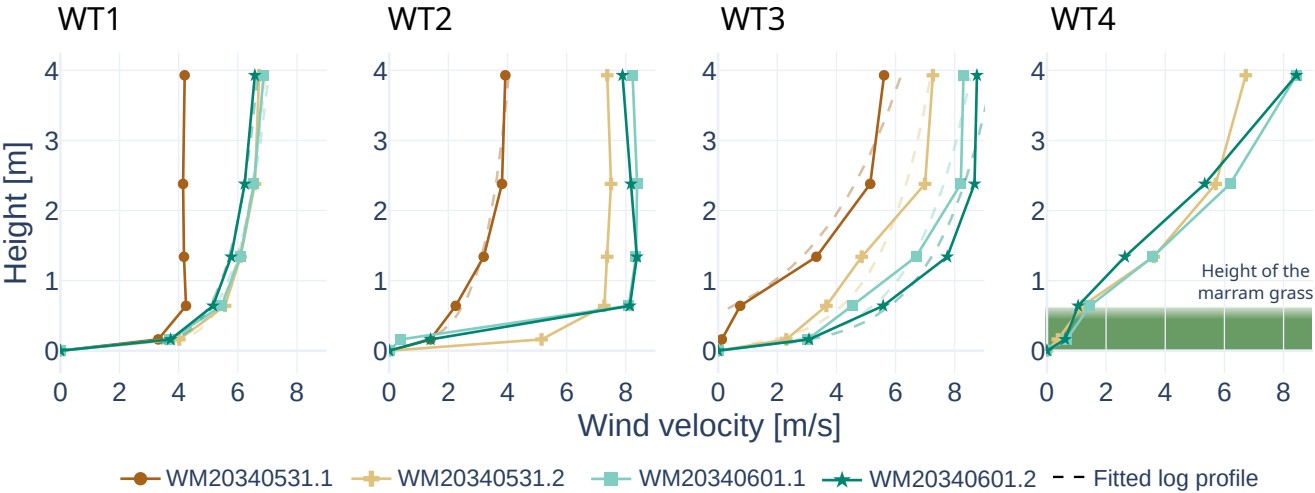

**Figure 4.** Representative measurements from the wind tower with location along the dune; log profiles fitted if feasible

Measurements of the aeolian sediment transport are named by the day of the month at the beginning of the measurement, followed by the day of the end of the measurement. A3101 started at 31.05.2023 in the evening and ended in the morning of the 01.06.2023. Three exemplary data samples of sediment transport measurements using the aeolian traps for positions AT1, AT4 and AT6 are shown in figure 5. The measured transport rates, $q$, are plotted over the elevation above ground; calculated rates of sediment transport $Q_S$ are additionally provided in the sub-figures as text. Extrapolation outside the measured area using





equation 2 is only feasible for measurements on the open beach at AT1, with A2930 being the only measurement providing

enough transport for an extrapolation. All other measurements did not result in an exponential decay function. For these, the

transport rate $Q_S$ is only calculated using linear interpolation and summation, using the vertical distributed data points of the

sediment transport flux $q$ with an additional zero crossing. Resulting transport rates are marked with a solid gray in figure 5.

The hatched area for the first trap is the surplus of the extrapolated calculation, it is cut off to zoom into the area with data

points. In total, $Q_{S,linear}$ is more than an order of magnitude lower than the extrapolated transport rate $Q_{S,extrapolation}$.

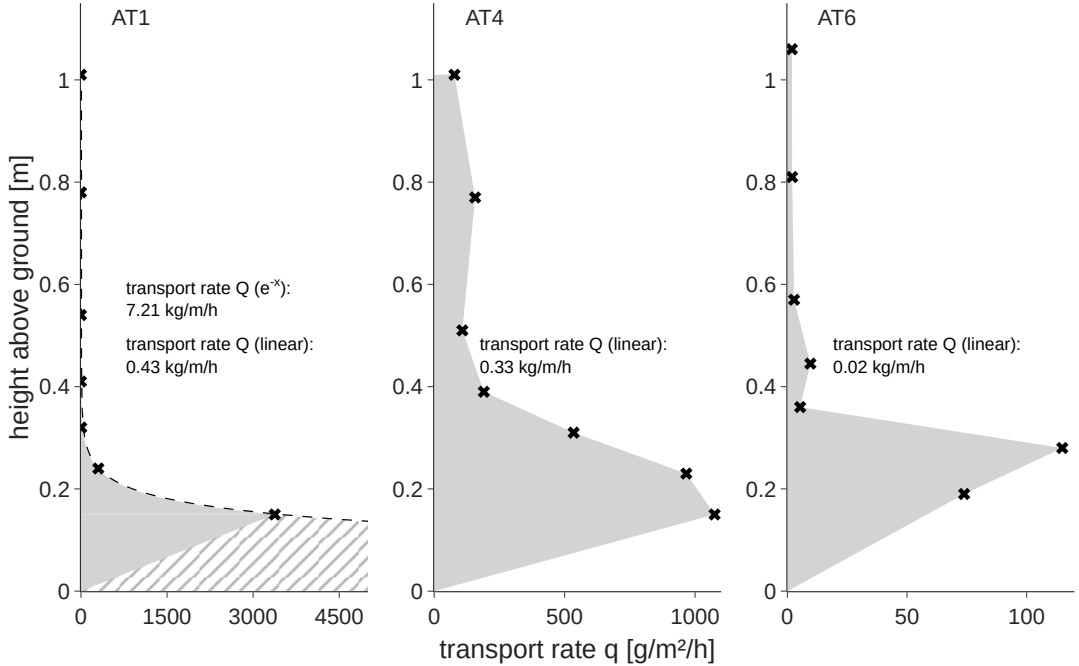

**Figure 5.** Representative examples of measured vertical distributions of the aeolian flux. The gray area is the transport derived via linear interpolation, with the hatched area being the additional transport gained using extrapolation.

The average wind velocity throughout the time window of the measurement A2930 was $\bar{u}_{10} \approx 6.52 \ m/s$. Using the law of the

wall with the average $z_0$ from table 1, this results in a shear velocity of $u_* = 0.296 \ m/s$. Inserting $u_*$ and $d_{50} = 0.2542 \ mm$

into the model by van Rijn and Strypsteen (2020) yields a transport rate $Q_{S,vanRijn} = 7.66 \ kg/m/h$, which is slightly higher

than the one from the extrapolated measurement of $Q_{S,extrapolation} = 7.21 \ kg/m/h$.

Because of the precipitation during the night from May 29th to May 30th, the upper layer of the surface sediment solidified

into a sand crust, residing on top of the dry powder sand underneath. The thickness of this crust ranged from roughly $0.5$ to

$1.5 \ cm$. Photographs of this layer can be found in the figures A4 to A7. This crust was not completely dissolved by the end of

the field trip three days later.



Figure 6 gives an overview of the wind directions and velocity compared to the measured transport rates. These rates are calculated using the linear interpolation ($Q_{S,linear}$). Each measurement is represented by a circle, for which its size and colour represent the amount of transport. The top row represents the first measurement, started at 17:20 on May 29th, with the last measurement at the bottom, started at 9:00 on June 1st. On the abscissa is the relative position to the foot of the dune, pointing northwards to the sea. Over the course of the week, there is no significant change in wind direction, whereas the wind velocity almost stops in the afternoon of May 30th.

Regarding the transport rates, there is a noticeable drop for the AT1 (positioned at $-20\ m$) after the first measurement A2930. The transport did not resume even after the wind velocity rose again on May 31st. Except for the outlier of the measurement A0101, AT2 measured almost no aeolian transport at all. When replacing the containers of the traps, it was visually observed that most of the transport at AT3 took place not around the inlets but in a small layer in between the individual BEST traps. The resulting transport rates are very low, with the last measurement A0101 as an exception. In comparison to AT1, the transport rates of AT4 to AT7 did resume with the rising wind velocities on May 31st. Even though AT7 is placed as closely as possible behind the tuft of marram grass, significant transport was measured for most measurements, but a continuous decline exists compared to the transport in front of this tuft at AT6. Three measurements of the traps AT4 and AT5 feature exceptionally transport rates.

Each measurement of the last two traps consistently shows lower transport rates compared to those discussed above, while the transport rate of AT9 consistently remains higher compared to AT8. Furthermore, no significant reduction in transport can be seen on May 30th, when the wind velocities almost stopped.

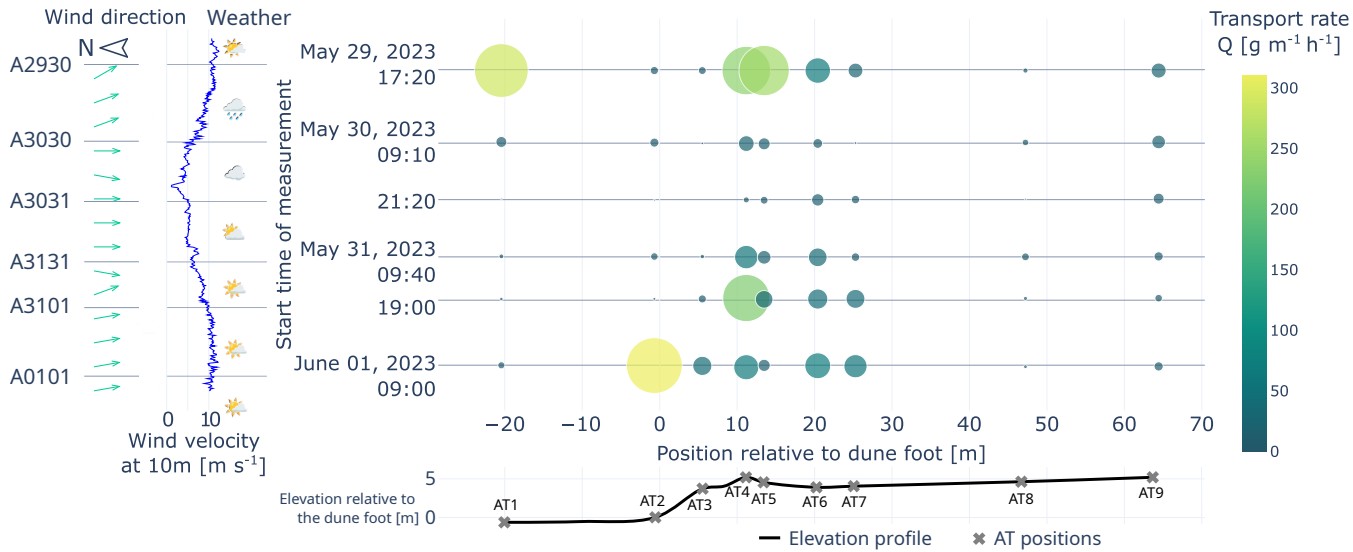

**Figure 6.** Measured aeolian transport rates from the field campaign compared to wind velocity and direction, cloud cover or precipitation and the position on the dune. Size and colour of the dots indicate the measured transport rates. The start time of each measurement is marked with dark gray lines.





## 3.2 From hourly transport rates to annual approximation of aeolian transport

We now aim to approximate the annual aeolian transported mass into the adjacent dunes of each beach section shown in figure 1, using widely available environmental data. To that end, it is necessary to quantify transport over the year based on a compilation of available meteorological data. As described in section 2.4, a time histories of shear velocity is required to

calculate annual transport rates. It is deduced by rearranging the law of the wall (equation 1), and inserting the time history of wind velocity $u_{10}$, measured by the DWD at a height of ten meters in ten minute intervals, and the zero roughness length $z_0$ of the study site from table 1. This work simplifying assumes that the zero roughness length remains constant over the course of a year. The respective aeolian transport volume is obtained by multiplying the transport rates of each interval by its duration of ten minutes.

Additional variables for the applied transport model are $\alpha_{dir}$ and $\alpha_W$. Both variables can reduce, or even prevent, the aeolian transport into the dune system. An annual distribution of these reduction factors is qualitatively shown in figure 7. All three beaches are covered by the time series of $\alpha_W$, but only the orientation of the northern beach is covered by the time series $\alpha_{dir,north}$. A northern wind direction ($\alpha_{dir,north}$ close to one), appears most often during the summer of 2022. The winter half-year shows fewer alignments in general and from October and December, the wind direction seldom aligns into the

northern dune system. The surface moisture, on the other hand, leads to a different distribution of $\alpha_W$. Overall, $\alpha_W$ limits transport less frequently than $\alpha_{dir,north}$. The frequency of a moist beach is higher in the fall and winter compared to the spring and summer, but only a few days at a time.

The bottom row highlights periods where both conditions align. In these periods, aeolian sediment transport into the northern dune system is possible. In comparison to the previous rows, it is notable that the distribution mostly resembles the $\alpha_{dir,north}$.

This similarity arises because the $\alpha_W$ only restricts transport briefly, leaving significant gaps where transport can occur.



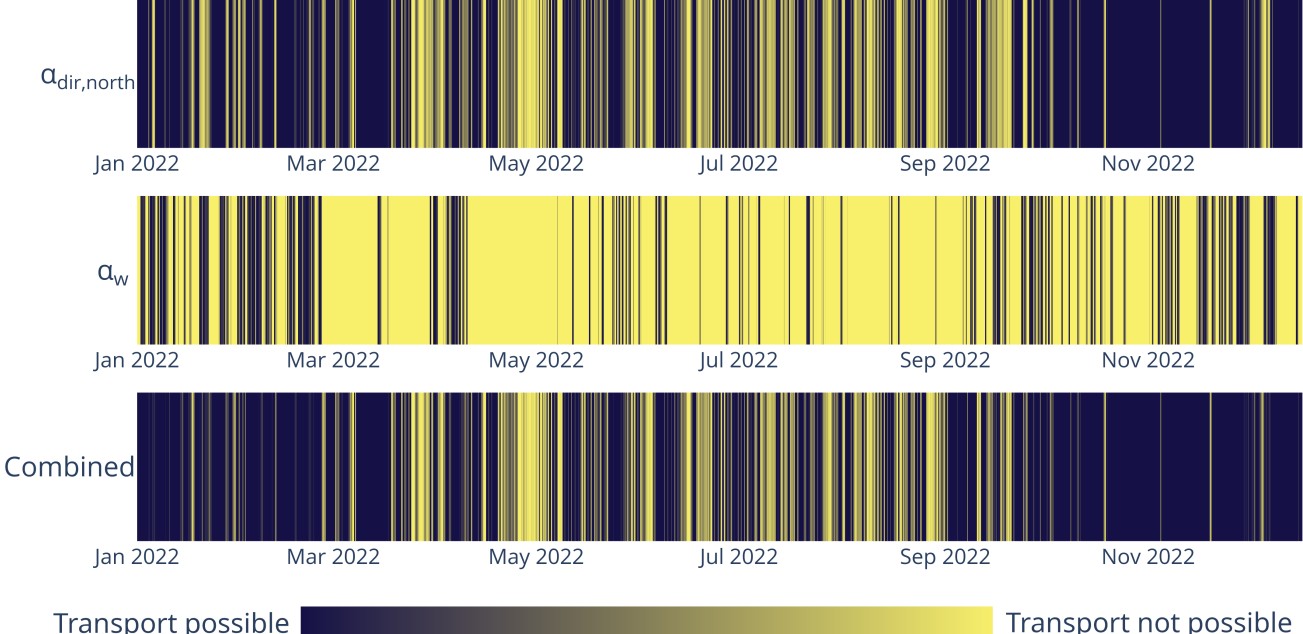

**Figure 7.** Distribution of the variables $\alpha_{dir,west}$, $\alpha_W$ and the combination of both for the year 2022. Transport is only possible if both variables align.

Table 2 lists is the annually aggregated aeolian transport for each of the three beach sections on Spiekeroog, further supplemented by the volume change for each beach and the adjacent dunes. The approximated annual transport into the dune system surpasses the actual dune growth by a third for the western beach. Moreover, the volume growth of the beaches itself surpasses the aeolian transport and dune growth by more than an order of magnitude. For the southern beach, the transport is around fifty percent above the annual dune growth and the beach exhibits the smallest volume grow up in comparison to the other two sections. What characterizes the northern beach is the discrepancy between the actual volume changes and the empirical approximation, which is roughly seven times larger. Similar to the western section, the volume changes of the beach also exceed the annual dune growth.





**Table 2.** Comparison of annual transported sediment on Spiekeroog per meter beach length between empirical approximation and volumes changes of the beach and dunes derived from geodetic scans

|  | Approximation | Dune | Beach |
| --- | --- | --- | --- |
|  | 2022 | Annual average 2018-2022 | |
|  | $[kg/m/y]$ | | |
| North | 33.13e+03 | 5.32e+03 | 41.43e+03 |
| West | 61.02e+03 | 45.59e+03 | 178.36e+03 |
| South | 30.60e+03 | 20.38e+03 | 12.81e+03 |

### 3.3 Deriving volume changes of the beach and dune systems from geodetic survey data

Contextualization is provided through federal survey data of the islands dune and beach area acquired in 2018, 2019 and 2022 using airborne Light detection and ranging (LIDAR) (REF). Data sets are available for the years 2018, 2019 and 2022 on a homogenized raster with a one-meter grid size. Consecutive years are subtracted from one another to calculate elevation changes and derive volume changes. The seaside boundary of the data set is delineated using the official mean tidal high water line. The corresponding land side boundary is defined along the grid cells, which exhibit a surface slope $> 5°$ corresponding

to the seaside dune toe along the island. This area is defined as upper beach width. The beach is segmented into three parts, based on their geographical orientation, yielding north *Nordstrand*, west *Weststrand* and south *Suedstrand*. Figure 8 shows cumulative sedimentation erosion values calculated for the period 2018 – 2019. Corresponding values are compiled in table 3. For the period from 2018 to 2019, an overall surplus in accretion is quantified for all sections. This finding is supported by official statements regarding beach and dune nourishment operations conducted by the Niedersächsischer Landesbetrieb für

Wasserwirtschaft, Küsten- und Naturschutz (NLWKN) to repair eroded dunes along the western and southern beach sections.

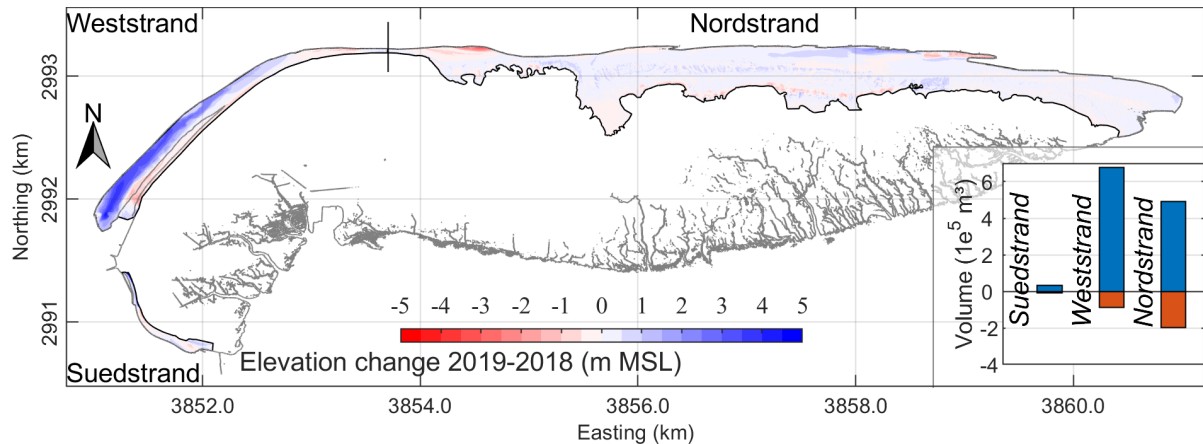

**Figure 8.** Absolute cumulative erosion accretion volume in $m^3$ for 2019–2018 based on federal survey data with 1m resolution.





**Table 3.** Geodetic survey data analysis for beach segments on the tidal barrier island Spiekeroog based on federal data for 2018, 2019 and 2022. Beach section data and corresponding cumulative erosion and accretion volumes are computed.

| Beach section | Length (m) | avg. width (m) | max. width (m) | min. width (m) | cum. erosion (m³) | cum. accretion (m³) |
|---|---|---|---|---|---|---|
| 2019 - 2018 | | | | | | |
| North | 7.3 km | 345 m | 825 m | 70 m | -198217 | 489832 |
| West | 3.3 km | 196 m | 370 m | 40 m | -89407 | 679180 |
| South | 1.2 km | 76 m | 127 m | 43 m | -10004 | 34245 |
| 2022 - 2019 | | | | | | |
| North | 7.3 km | 346 m | 826 m | 68 m | -192693 | 417642 |
| West | 3.3 km | 195 m | 368 m | 38 m | -34144 | 405158 |
| South | 1.2 km | 69 m | 126 m | 42 m | -26467 | 9674 |

For the ensuing period from 2019 to 2022, no intermediate data sets are available, thus a cumulative change over a period of three years is calculated by subtracting 2019 bathymetry data from the 2022 data set resulting in figure 9. Corresponding data is compiled in the bottom part of table 3. For the section *Nordstrand*, a comparative erosion and sedimentation compared to the period 2019-2018 is calculated. For the western beach section, smaller volume changes are computed for the extended period 370 of time. However, for the southern beach section, a comparatively large erosion volume is quantified.

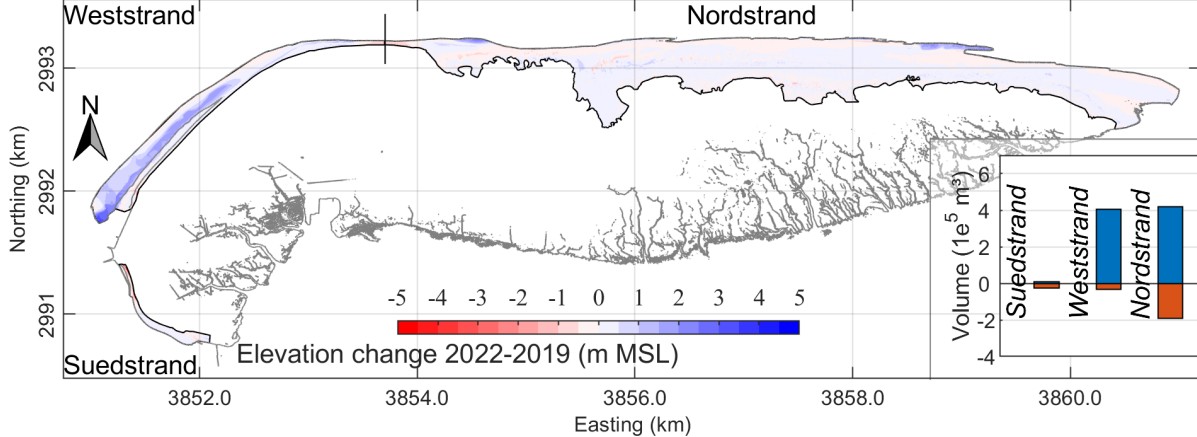

**Figure 9.** Absolute cumulative erosion accretion volume in m³ for 2022 - 2019 based in federal survey data with 1m resolution.

Adjacent dune systems corresponding to the three beach sections defined were analysed, starting at the beach polygon boundary, shown in figure 1, and ending after the first major crest to include the frontal dune system, directly impacted by aeolian sand transport and wave induced erosion.





**Table 4.** Geodetic survey data analysis for dune system segments on the tidal barrier island Spiekeroog based on federal data for 2018, 2019 and 2022.

| Dune section | area ($m^2$) | average max (m) | dune height (m) | cum. erosion ($m^3$) | cum. accretion ($m^3$) |
|---|---|---|---|---|---|
| 2018 | | | | | |
| North | 1369646 | 3.66 | 14.59 | -28850 | 74762 |
| West | 288054 | 10.78 | 20.37 | -53180 | 218628 |
| South | 65978 | 5.58 | 14.97 | -3609 | 33362 |
| 2019 | | | | | |
| North | - | 3.83 | 15.21 | -28850 | 74762 |
| West | - | 10.96 | 20.65 | -53180 | 218628 |
| South | - | 5.91 | 15.12 | -3609 | 33362 |
| 2022 | | | | | |
| North | - | 3.91 | 15.21 | -43032.92 | 46530.80 |
| West | - | 10.97 | 20.63 | -84506.47 | 135349.95 |
| South | - | 5.93 | 15.1 | -9545.15 | 9230.57 |

# 4 Discussion

## 4.1 Field study

Quantification of aeolian transport rates on top dunes proves to be challenging. The issue with sediment traps, which only measure at discrete vertical points like Modified Wilson and Cooke (MWAC) or BEST traps, is the requirement of interpolation within the measured area and extrapolation beyond it. Transport which only occurs in between two inlets is not getting measured, as described in section 3.1 for AT3, and therefore does not appear in the derived transport rates. This neglection also applies to transport below the measured area, which can be assumed to be the case for AT4 in figure 5. In these cases, an extrapolation outside the measurements is not feasible due to the unknown vertical transport distribution. This vertical distribution of the transport rates will vary between the positions of the aeolian traps used in this study due to the impact of neighbouring vegetation and surface structure. As a result, the deviation between the measured transport rates and the actual ones is influenced by the chosen location. Traps that measure vertically continuously, such as the trap from Sherman et al. (2014) or Rotnicka (2013), solve this problem of transport neglection on the one hand, but on the other hand, do not align themselves with the wind direction, which could have an impact on real world efficiency.

Another error source to be considered are visitors interested in the measuring equipment. These can stir up the sediment surrounding the traps, resulting in invalid data points. This could be an explanation for the outliers at positions two, three, and four in figure 6.





The only measurement, which led to a quantifiable transport rate with a probably good accuracy, was the measurement of AT1 on May 29th. Even though, the transport should have resumed after the low wind velocities on May 30th, almost no transport was measured. One significant reason is probably the solidified surface crust on the beach, that leads to a lack of loose sediment which could have been transported by the wind. For no clear reasons, the AT2 did not measure any significant transport, despite its location in front of the dune. Here, the dune geometry could already influence the wind field and actively disturb the vertical

distribution normally found on the open beach. This phenomenon is described as aeolian ramp in Rotnicka et al. (2023). Unlike the measurement taken on the open beach, the transportation atop the dune (AT4 to AT7) seems unaffected by the crust on the open beach because the measured transport rates increased with the wind velocity on May 31st. The lack of transport on the open beach means that no new sediment is transported into the dune in high quantities, which in turn implies that most of the transported sediment has to originate from the dune itself. On the rear side of the dune (AT8 and AT9), no effects of the lower

wind speeds can be observed. The overall low transport rates could mean that the sediment has already settled in the front area of the dune.

The application of the model by van Rijn and Strypsteen (2020) for the night from May 29th to May 30th resulted in a good fit to the measured transport rate on the open beach. In turn, this means that the combination of zero roughness length derived from the wind profile measurements on the open beach and the measurement of the wind velocity at $10\,m$ by the DWD delivers

a valid shear velocity for this model.

Measured wind profiles of WT3 (next to AT6) show a significant wind velocity close to the ground, which means that the sediment is set in motion at this position. This stands in contrast to measurements at WT4, located behind a tuft of marram grass. Here, the lower $0.8\,m$ show low wind velocities, which should result in settling sediment.

## 4.2 Approximation

As expected, the approximated annual aeolian transport of sediment into the dune system surpasses the volume changes of the dunes for each analysed beach section. This is necessary because the settling sediment must initially replenish the volume eroded during storm surges before the dunes can grow. On the western section of the beach, the estimate exceeds the annual dune growth by approximately thirty percent. If one assumes that all sediment transported into the dunes is also deposited there, this implies that storm surges erode this thirty percent annually, equating to roughly 15 tons of sand per meter of length. In

contrast, the northern beach exhibits an aeolian transport exceeding the dune growth by a factor over six, which is significantly higher compared to the western beach. One explanation could be, that the dune system exhibits a higher erosion during storm surges, which thereby has to be rebuilt by the aeolian transport. Compared to the other beaches, the southern section is narrower. A mean width of $69\,m$ suggests that the aeolian transport could often be limited by a short fetch length. Such limitations are not considered this approximation. Nourishments of the beach and dune are also included in the volume changes, which hinders a

comparison to the approximated transport.

Another influence on the approximation is the distance of the beaches to the meteorological stations. These stations by the ICBM and DWD are positioned on top of the dunes adjacent to the western beach, meaning that the measured meteorological



conditions could differ from the ones found on the beaches. However, on an annual scale, the meteorological conditions should be roughly equivalent over the whole island, meaning that any short-term difference should be negligible when summing up

the whole year. Another source of inaccuracies in the approximation lies in the assumption, that $z_0$ is a constant, whereas in reality it is varying on an annual scale, as described by Strypsteen (2023).



## 5 Conclusion

This paper presents the results of a field study measuring the aeolian transport and wind profile across a dune and an approximation of annual transported sediment into the dune systems of the east frisian island Spiekeroog based on widely available
meteorological data.

Objectives outlined in section 1 comprise the conduction of episodic, local aeolian transport and wind profile measurements for dune faces with and without vegetation canopy. This was partially accomplished, in that field data was successfully acquired on baren dune faces but not within vegetated areas. This is mainly attributed to the types of traps used, which do not cover the whole vertical range but only dedicated heights. The acquired data was correlated with regional long-term weather data for
multiple years. Aeolian sediment fluxes were successfully computed for beach-dune sections. Furthermore, sensitivity of the transport volumes towards precipitation, sunshine duration and intensity has been tested and quantified limiting transport rates during rain events and subsequently increasing them again given the sunshine duration. Volume changes in beach and dune sections have been computed for multiple years based on federal digital terrain models. Computed volume differences based on the digital terrain models are compared with projected transport volumes based on long term weather information informed
by conducted wind and transport measurements. The volume flux projections are upscaled from local measurements to a whole barrier island.

Due to the partial neglection of aeolian transport caused by the trap design used in this study, actual transport rates are very likely higher, and therefore a reliable quantification of the transport rates inside of vegetation is not possible. However, it was qualitatively shown that transport is still possible, even behind $20\,m$ of vegetation on top of a dune, which corroborates findings
reported by Rotnicka et al. (2023). Actual transport rates are very likely to be higher than measured, given aeolian sediment transport trap efficiencies reported (Sherman et al., 2014; Eichmanns et al., 2021; Basaran et al., 2011; Horikawa and Shen, 1960). This transport eventually settles, as the two aeolian traps positioned furthest into the dune did not measure significant transport. To investigate the quantity and exact pattern of sediment movement and settling along a dune, sediment traps that assess more than just specific vertical points would be essential, suggesting a design similar to that reported by Rotnicka
et al. (2023). Transport on the open beach paused after light precipitation for multiple days due to solidification of the upper sediment layer. Such transport reductions will influence the volume of sediment transported into the dunes on an annual scale. Accordingly, the sensitivity of the aeolian transport was investigated by means of reducing the transport depending on the local weather observation data. Calculations conducted within this study clearly show the impact it has. Henceforth, transport is highly sensitive to precipitation and sunshine duration, dedicating the overall sediment humidity (Homberger et al., 2024).
Underlined is this observation by the measurement of fairly high wind velocities near the ground at WT3. This suggests that sediment will continue to move further into the dunes, despite the occasional absence of new material transport from the beach, which is controversial at this point, as recent findings support (Rotnicka et al., 2023) but also contradict this finding (van Rijn and Strypsteen, 2020).



Analysis of geospatial data from 2018, 2019 and 2022 shows, that the beaches and adjoining dunes of the Island Spiekeroog are growing, for the most part, even without any nourishment. The application of the transport model by van Rijn and Strypsteen (2020) on an annual scale delivered transport rates, which exceed the actual volume changes derived from the geospatial data, which is to be expected. What stands out is that the difference in between annual transport and volume changes of the adjacent dune system for the northern beach is significantly higher than that of the western beach. It would be beneficial to record geospatial data after the winter storm surges and again directly before surges in autumn. Volume changes derived from this data should be roughly matched by an approximation of transported sediment in this time window.

*Code availability.* Scripts developed within this research study can be made available upon reasonable request by the authors

*Data availability.* Fielddata acquired within this research study can be made available upon reasonable request by the authors. Federal data used within this study is publicly available.

*Sample availability.* Data on acquired field sample data can be made available upon reasonable request by the authors.

*Video supplement.* There are no video supplements.





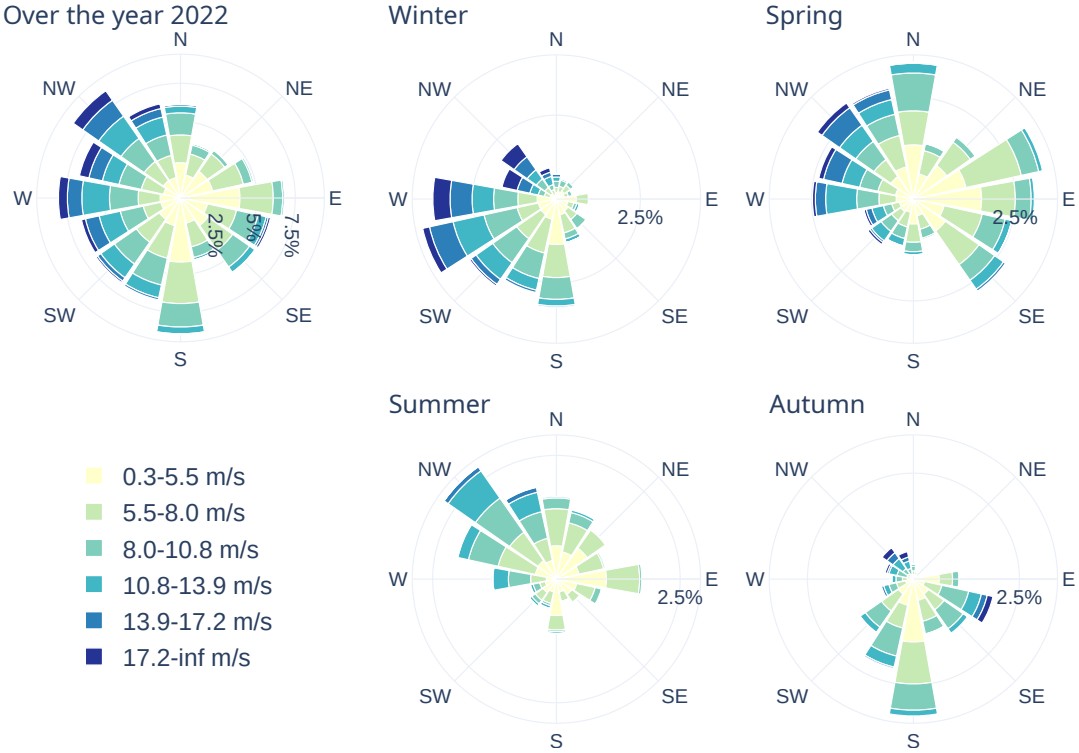

**Figure A1.** Wind roses for the entire year and each season in 2022.



(a)

(b)

(c)

**Figure A2.** (a) Arrangement of components on the mobile wind tower, (b) dimension of the aeolian trap, (c) Dimensions of the BEST trap (Basaran et al., 2016).





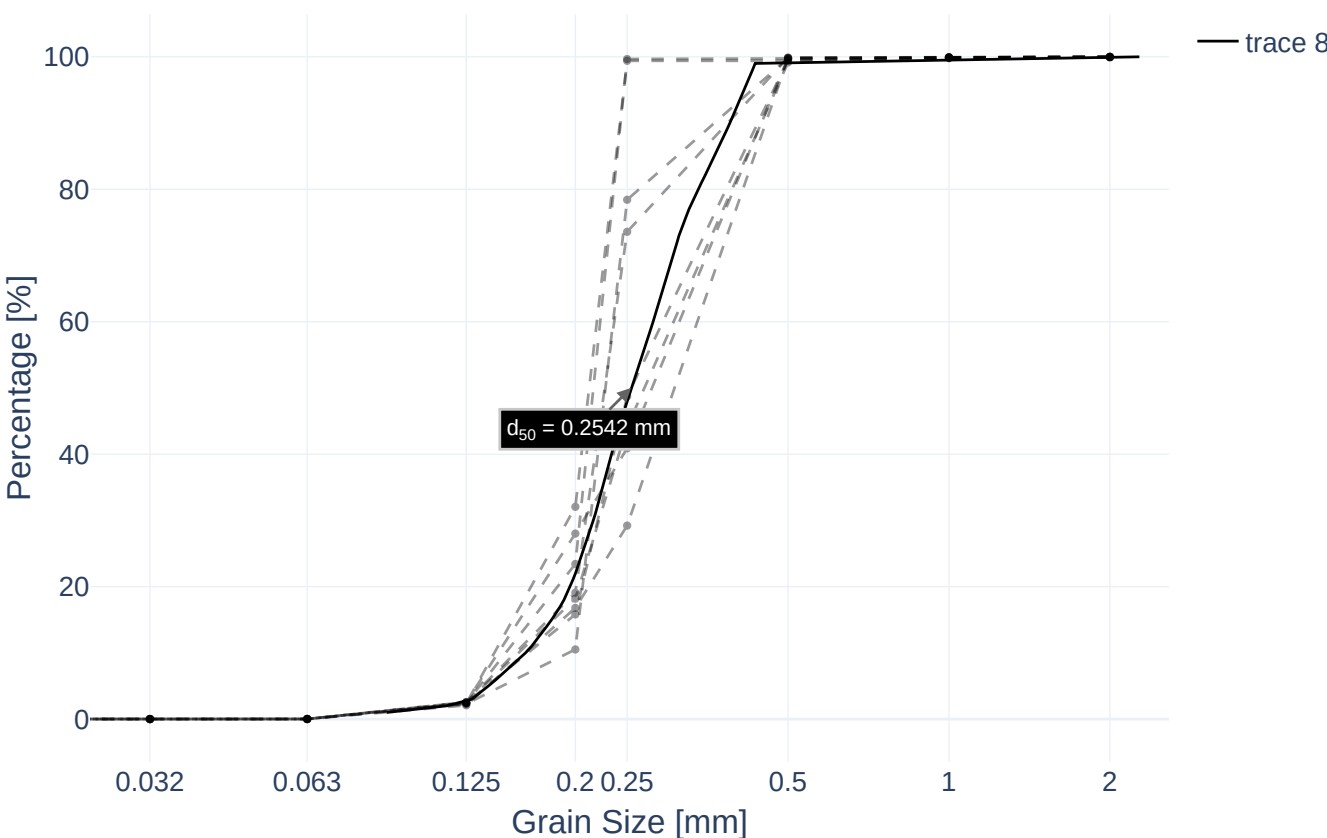

**Figure A3.** Sieve curves of sediment probes (dashed) and mean sieve curve in black with marked $d_{50}$





**Figure A4.** The crust formed a continuous layer with dry powder sand beneath





**Figure A5.** The thickness of the crust formed due to precipitation was below one centimetre





**Figure A6.** Stable peace of the crust which could be lifted and did not fall apart on its own



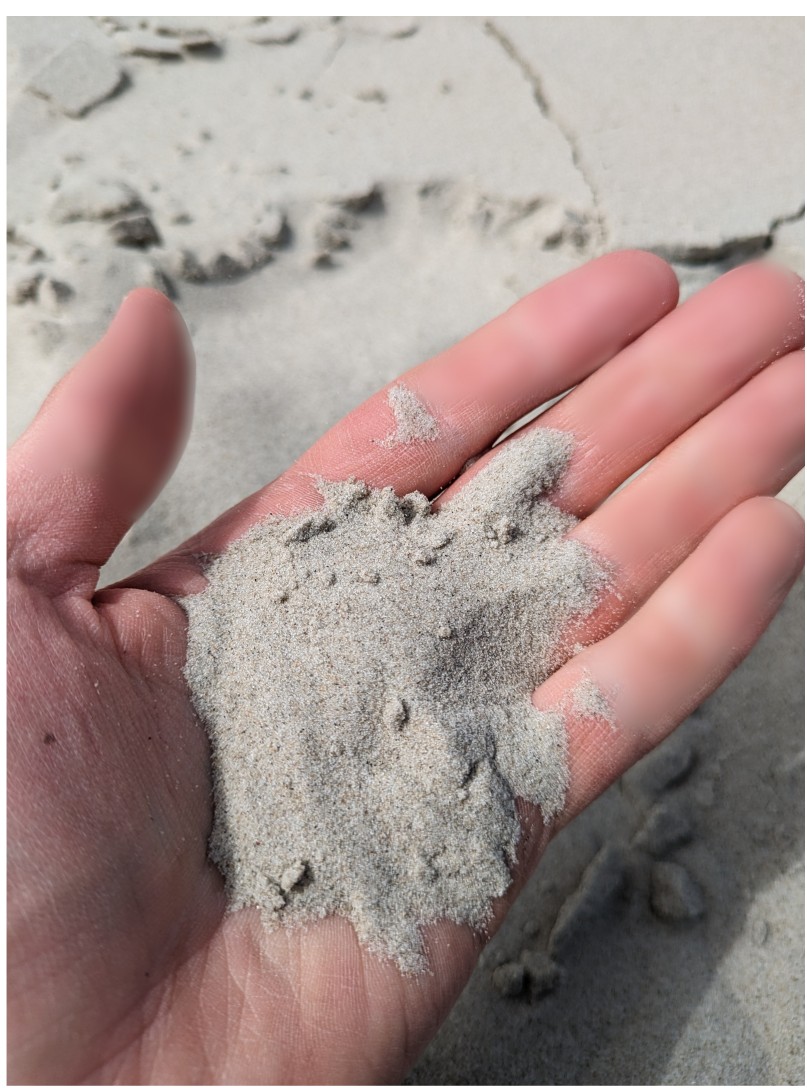

**Figure A7.** Crushed peace of the crust turned into dry powder sand



**Table A1.** Time schedule of the wind profile measurements

| Measurement | Position | Date | Time | Title string |
|---|---|---|---|---|
| 1 | 1 | 31.05. | 10:53 | Wednesday Morning |
| 1 | 3 | 31.05. | 11:38 | Wednesday Morning |
| 1 | 6 | 31.05. | 12:20 | Wednesday Morning |
| 1 | 7 | 31.05. | ERROR | Wednesday Morning |
| 2 | 7 | 31.05. | 16:04 | Wednesday Evening |
| 2 | 6 | 31.05. | 16:46 | Wednesday Evening |
| 2 | 3 | 31.05. | 17:30 | Wednesday Evening |
| 2 | 1 | 31.05. | 18:11 | Wednesday Evening |
| 3 | 1 | 01.06. | 09:05 | Thursday Morning |
| 3 | 3 | 01.06. | 09:45 | Thursday Morning |
| 3 | 6 | 01.06. | 10:22 | Thursday Morning |
| 3 | 7 | 01.06. | 11:15 | Thursday Morning |
| 4 | 7 | 01.06. | 14:45 | Thursday Evening |
| 4 | 6 | 01.06. | 15:28 | Thursday Evening |
| 4 | 3 | 01.06. | 16:07 | Thursday Evening |
| 4 | 1 | 01.06. | 16:47 | Thursday Evening |

*Author contributions.* MK: Manuscript draft writing, sensor development and assembly, field measurements, field data evaluation, transport calculations, figures; OL: Manuscript draft writing, editing, measurement conceptualization, figures, funding acquisition; VK: Editing, sensor assembly; BM: Editing, field measurements; LA: Editing; DS: Editing, funding acquisition; NG: Editing, funding acquisition

*Competing interests.* There are no competing interests to the authors knowledge

*Acknowledgements.* The research presented within this study has been funded by the Ministerium für Wissenschaft und Kultur Niedersachsens and the Volkwagen Stiftung in equal parts under ther Gute Küste Niedersachsen grant FKZ: Gute Küste Niedersachsen is funded by the Lower-Saxony Ministry of Research and Culture (FKZ: 76251-17-5/19) and the Volkswagen Stiftung. The authors would like to extend their



gratitude to the Wadden Sea National Park for issuing the field measurement permits to acquire field data within a national park. Finally, the authors extend their thanks to Holger Dirks from the Forschungsstelle Küste of the Niedersächsischer Landesbetrieb für Wasserwirtschaft, Küsten- und Naturschutz for providing high resolution digital elevation model data for this study.



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
