# Peer review of "Measurements of aeolian sediment transport in the vicinity of coastal dunes on Spiekeroog Island, Germany, and extrapolation to annual transport volume"

_EGUsphere, 2024_

## Referee Comment (RC2)

The issue of using short-term measurements of aeolian sand transport rate on the beach to estimate the annual potential transport and then compare it with the actual sediment budget of coastal dunes is still one of the biggest challenge. This issue has been addressed by many researchers for a long time, but probably much research is still needed to reach any consensus. Therefore the submitted manuscript fits very well into this research trend. The objectives of the paper (listed in lines 124-129) are clearly stated but they are so extensive that each of them could be a separate research task. If we do not treat them properly, the results obtained will be random and will not expand our knowledge. I appreciate the authors' intentions, but they probably did not realize the complexity of the issue.

I have many comments and remarks to the manuscript, which make me not recommend it for publication as it is now.

Comment 1: STUDY SITES lacks description of different beach segments of Spiekeroog (why was the island divided into 3 sections?), METHODS and experiment procedure are described superficially. CONCLUSIONS are not supported with data. The manuscript is written chaotically. For example, Section 3 (Results) and subsection 3.2 - First, wind profiles are described, then sand transport, and eventually the conditions in which the sand transport intensity was measured (but they are probably attributed to the speed data from the weather station located somewhere, not from authors' measurements made at the study site). Wind direction, which is very important factor when it comes to determining the available fetch distance and airflow forcing and steering is described at the end of this section. .

Comment 2: Sand transport rate was measured by BESTs traps and only at one site the vertical distribution of mass flux was described by an exponential decay function. At other sites, however, the vertical distribution of mass flux was not known (??, line 212) and the authors used linear interpolation of point data. There are many functions used to describe the vertical distributions of aeolian mass flux (as exponential, logarithmic, power, and – on rough surfaces – Gaussian or a combination of different functions), but no one uses a linear function. If the authors use such a trend, they should justify it. Which entitles them to calculate the total transport intensity based on such a trend? Further on (lines 295-299), the total sand transport is calculated on the basis of a linear trend, but it is additionally cut off close to the ground where in fact the sand transport rate is the highest, provided it is not within a vegetation. I do not understand this idea. Additionally, the annual potential sand transport was calculated from Van Rijen and Strypsteen (2020) model, using the data from nearby (?) weather station and assuming a priori that it reflects given conditions and studied beach-dune system well. Its outcome was verified by a comparison with the result of a single measurement of the sand transport rate (lines 300-304).

Comment 3: The measurements of sand transport rate were made during four days at relatively low wind speed (looking at Fig. 5 it was probably less than 6 m/s at 1 m height, site on the beach), so the sand transport must have been relatively weak and probably intermittent. One of the measurements was carried out throughout the night (lines 289-290), so if the BEST trap did not fill up with sand for such a long time, the transport must have occurred occasionally. Some of the measurements were accompanied by rain which probably suppressed such a weak transport to zero. The maximum sand transport rate was 300 g/m/h (Fig. 6) – it means 3 g of sand per 1 cm of beach during 1 hour. My conclusion is that the data set collected by the author does not represent sand transport conditions on the beach and cannot be used as a basis for any further calculations or model testing.  And in fact they are not used further on.

Comment 4: The authors compare the annual sand transport rate calculated on the basis of a model developed by Van Rijen (2018) and Van Rijen and Strypsteen (2020) with transport rate derived

(how?) from the geodetic data (volume changes of beach and dune), but in Chapter 4.2 we got to know that the beach and dune was nourished with sand!!

In fact, the measurement of sand transport rate presented in the manuscript are of low value, but they are unnecessary for the rest of the paper. Therefore I suggest to remove this part and better organize the article.

**Summing up, the submitted manuscript needs major revision and should be review again.**

Some more remarks and comments:

Lines 80-81 "the available fetch length between the tidal high water line and the dune foot, usually in the form of a beach"? Can it be anything other than a beach?

Line 95 – ebda?

Line 97 – not to "to lateral upwind canopy coverage percentages" but to density of vegetation.

Lines 98-99 – ranges of vegetation density – cite the source of this data

Line 148 – What do you mean by "primary dune row"? A foredune?

Lines 158-159 – the first sentence is unnecessary

Lines 171-174 – it is not clear. You used only BEST traps, didn't you?

Line 187 – you should provide at least an approximate grass density and height as depending on it sand transport is different

Line 191 – "In total, six data sampling intervals lasting each half a day were logged". You used BEST traps – did they operate for half a day?

Table 1 – are these values averages calculated for wind event during experiments? What was the actual wind speed at any elevation and what was its variability during the measurements?

Line 212 – you did the research, not Williams (1964)

Equation 4 – there is no explanation of all symbols used in the equation: $Q_D$, $\alpha_D$, $u_{*,th}$ Some of them are explained further in the text, but it is too far

Line 221 – median, not medium. How did you determine the critical fetch distance?

Line 229 – incipient velocity?

Line 234 – replace solid with grain or particle

Line 238 – what do you mean by "nearby weather station" Explain in the section on Study site.

Line 248 – air humidity

Line 284 – what is front crest?

Fig5 – use the same unit for q and Q (either grams or kilograms) here and throughout the text

Line 300 – do these wind data come from your measurements or from the weather station which is far from your study site?

Line 335 – explain $\alpha_{dir}$ and $\alpha_W$

Line 345-350 and Table 2 – you provided calculated sand transport rate for 3 beach sections, but they are not described in the study site. Why is Spiekeroog divided into 3 segments, what are the differences between them?

Table 2 – how did you calculate annual sand transport rate on the basis of geodetic data? You can calculate volume changes, but how did you transform it into the transport rate. Did you assume any sand bulk density? Provide it. There is not any explanations in METHODS.

Lines 393-395 – "For no clear reasons, the AT2 did not measure any significant transport". This site was at the base of a dune, were the airflow decelerates due to an increase in the pressure gradient (during onshore or oblique onshore winds). Analyse the data on wind direction! I can only guess that it was alongshore or oblique onshore (Figure 6). Additionally, the wind during the measurements was very weak, it's no wonder that sand was not transported there. Whether the sand transport will occur here or not depends on both the wind force and direction.

Conclusions – this section repeats what has been done (lines 430-440), and further on there is a kind of a discussion with literature (lines 442-459). In fact these are not conclusions. In lines 459-460 the authors stated that dunes grow that independently of their nourishment but I do not any information on the magnitude of the nourishment and therefore I do not know how the authors came to this conclusion.

---

## Author Comment (AC1)

**Response to the reviewers**

Dear editor and reviewers,

The authors appreciate the time and effort invested by you and the reviewers in improving this manuscript. We believe that addressing the feedback has significantly enhanced its quality.

Please find attached a marked version of the manuscript, indicating the amendments the authors performed. Detailed answers to the reviewers' comments are provided in this rebuttal letter.

Answers to the individual comments can be found below. Line numbers refer to the position in the initial version of this manuscript. The convention is as follows:

- Refers to the reviewer's comments

- *Refers to our responses to the comments*

- Refers to new text added to the manuscript

-

*Reviewer 1*

*Major comments*

- Figure 5 – I would suggest standardizing the x-axis across the plots for AT1, AT4, and AT6. Right now, the transport rates look misleading as if AT4 and AT6 have similar rates between 0.2-0.4 m, yet they are a magnitude different.

  Response: *Thank you pointing this out. We agree that the usage of equal x-axis ranges was misleading. We have adjusted the ranges for the two right-hand subplots, but retained the larger range for the left-hand subplot, as the measured transport rate is a power of ten higher. However, we have changed the width of the sections to create a distinction between the left-hand x-axis range and the other two x-axes, which now feature equal ranges. If all three subplots feature the same x-axis ranges and widths, the shape of the vertical distribution of the two left measurements would not be recognizable.*

[Figure]

Figure 5. Representative examples of measured vertical distributions of the aeolian flux. The gray area is the transport derived via linear interpolation, with the hatched area being the additional transport gained using extrapolation.

Figure 2: Enter Caption

[Figure]

Figure 1: Representative examples of measured vertical distributions of the aeolian flux. The gray area is the transport derived via linear interpolation, with the hatched area being the additional transport gained using extrapolation.

- Discussion – The discussion introduces challenges faced in the field, and how these challenges (alongside assumptions) could be responsible for variability throughout the instrument. **I believe the discussion could be lengthened and serve to relate the findings introduced her back into the literature more heavily, which was explored further in the Introduction.** (e.g., Relate to other field experiments exploring wind flow over a dune, scarp, vegetated dune, etc.) Even if many experiments are at the

"event/episodic scale" across the literature, they would be applicable for this discussion. **Further, I would restructure the conclusion to move relativity to previous experiments/papers to the discussion** (right now the conclusion feels more like a discussion than a summary of main findings).

Response: *Tank you for this comment. The authors agree and we have revised the conclusion.*

- Line 304 – How much precipitation? Consider adding time series of local rainfall into Figure 6.

  Response: *Thank you for this comment. We agree, that a time series of precipitation would be benificial. However, the meteorological station on Spiekeroog from the ICBM, which measures precipitation, was faulted during the field campaign. The next station measuring the precipitation is positioned on the island Norderney, about 35 km away from Spiekeroog. Due to the relatively localized occurrence of rain on an episodic scale and the episodic nature of field measurements, the usage of data from Norderney would be questionable for a comparison here from out point of view. To nevertheless convey the general weather conditions during the field measurements, icons qualitavely describe the precipitation and cloud cover.*

- Line 436 – To quantify "sunshine duration" I would advise to include a time series for precipitation, especially given the relevance for surface crusting and resulting impact on aeolian transport potential. This is of critical importance when estimating yearly transport given local meteorological conditions.

  Response: *Thank you for this comment. The authors agree that the precipitation is important for this. A time series of the precipitation is not included, as only precipitation up to 0.3 mm are relevant for each event due to the field capacity. When including this, the peaks are way above 0.3 mm, and would overshadow smaller, but importand events. The shown time series if $\alpha_w$ gives an overview over the time periods with precipitation.*

*Technical Corrections*

Technical Corrections:

- Line 19 – Is that the correct format for the Bagnold reference?

  Thank you for pointing this out. We have corrected the name in our bibtex file.

   Bagnold, 1937

- Line 67 – von Karman constant; check spelling

  Response: *Thank you for bringing this to our attention.*

  von Kármán

- Line 68 – check formatting on Nikuradse reference, I believe it should be "by Nikuradse (1931),"

  Response: *The authors agree. We have removed the brackets*

  Nikuradse (1931)

- Line 111 – Sentence beginning with "Jackson and Nordstrom (2011)" is a bit confusing. Consider reframing or splitting sentence.

  Jackson and Nordstrom (2011) reviewed dune management methods and highlighted the importance of deepening knowledge of quantifying aeolian transport and the impacts that man-made structures or destruction through vegetation trampling have on it.

- Line 151 – Remove italics from "1.5 m" for consistency.

  Response: *Thank you very much for drawing that to our attention.*

  m

- Line 153 – check for consistency throughout; use of space between value and unit (9.45m or 9.45 m)

  Response: *Thank you for pointing this out.*

- Line 156 – Be consistent with units/degrees of measurement. Should be 3.50 to 12.00 m following text above which provides measurement down to the ten's unit.

  Response: *This is a valuable observation, thank you.*

- Line 166 – Check consistency with units (italics or not)

  Response: *Thank you for pointing this out, we removed italics from the units throughout the document.*

- Line 167 – maintain units (0.2 to 4.0).

  Response: *We appreciate you bringing this to our attention. We have corrected this throughout the document.*

- Line 180 – Replace extents with extends.

  Response: *Thank you for noticing this error.*

  extends

- Line 190 – Consider quantifying "half a day" into hours of data recording.

  Response: *The authors agree, that this sentence is too broad. The exact time were added in the appended table A1.*

  It was aimed to measure durations of half-day cycles, with the exact intervals listed in table A1.

- Line 194 – Incomplete sentence "The resulting shear velocities...". Refinement needed.

  Response: *We are sorry that this sentence was included. The paragraph was adjusted. Thank you very much for pointing this out.*

  The resulting shear velocities and zero roughness length are compiled in table 1 with the time windows presented in table A2 and positions in figure 3.

- Line 321 – Grammatical error – exceptionally (should this be exceptional?)

  Response: *Thank you for this correction.*

   exceptional

- Line 332 – Grammatical error "This work simplifying assumes"; consider rewording start of sentence.

  Response: *Thank you for pointing this out.*

  This work uses the simplification that the zero roughness length remains constant over the course of a year.

- Line 343 – Confusion on where "The bottom row" is referencing. Please include figure and panel reference.

Response: *Thank you very much for this addition. We have added lettering to the individual rows.*

The combination of both variables is shown in figure 7c. In periods where both reduction factors are not hindering, aeolian sediment transport into the northern dune system is possible.

- Line 346 – Grammatical error: "Table 2 lists is the annually" – Remove "is".

  Response: *I appreciate you pointing this out.*

- Line 350 – Consider replacing "grow up" with "increase"

  Response: *Thank you for this suggestion, we have replaced it.*

  increase

- Line 356 – Add reference to LIDAR.

  Response:

  Contextualization is provided through federal survey data of the island's dune and beach area acquired in 2018, 2019 and 2022 using airborne *LIDAR* derived *DEM* (GDWS, 2021; NLWKN, 2023; GDWS, 2024).

- Table 3 – remove units from values throughout table; units are defined in top row.

  Response: *Thank you noticing this douplication. We have removed the units from within the table.*

- Line 367 – Bathymetry data, or topographic LIDAR?

  Response: *Thank you for pointing out this inaccuracy, we have corrected it (see comment above for Line 356)*

- Line 415 – "aeolian transport [rate] exceeding the dune growth"

  rate

- Line 416 – Remove comma after "could be"

  Response: *Thank you for this tip.*

- Line 419 – "not considered [with/in] this approximation."

  Response: *Thank you for this addition.*

  with

- Line 438 – Consider checking for consistency throughout on the reference for federal LiDAR DEMs. It seemed to be referenced differently throughout the manuscript which can lead to confusion (LiDAR, bathymetry, terrain models). Include reference to the dataset each time it is referenced.

  Response: *Thank you for bringing this up. We have clarified the type of scans throughout the manuscript.*

*Reviewer 2*

*Major comments*

- STUDY SITES lacks description of different beach segments of Spiekeroog (why was the island divided into 3 sections?), METHODS and experiment procedure are described superficially. CONCLUSIONS are not supported with data. The manuscript is written chaotically. For example, Section 3 (Results) and subsection 3.2 - First, wind profiles are described, then sand transport, and eventually the conditions in which the sand transport intensity was measured (but they are probably attributed to the speed data from the weather station located somewhere, not from authors' measurements made at the study site). Wind direction, which is very important factor when it comes to determining the available fetch distance and airflow forcing and steering is described at the end of this section.

  Response: *We are grateful to you for identifying these areas for improvement, which we have addressed as follows:*

  *In the description of the study site, we included the beach segments, with an explanation for splitting them due to their orientation and included the locations of the meteorological stations.*

  *Further, we deepended the description of the methodology of the annual approximation and the execution and analysis of the field study.*

  *The conclusion has been rewritten with a focus on refering to the data presented in the results.*

*Additionally, we restructured the sequence for presenting the methodology and result section to address the structure of the manuscript.*

- **Comment 2:** Sand transport rate was measured by BESTs traps and only at one site the vertical distribution of mass flux was described by an exponential decay function. At other sites, however, the vertical distribution of mass flux was not known (??, line 212) and the authors used linear interpolation of point data. There are many functions used to describe the vertical distributions of aeolian mass flux (as exponential, logarithmic, power, and – on rough surfaces – Gaussian or a combination of different functions), but no one uses a linear function. If the authors use such a trend, they should justify it. Which entitles them to calculate the total transport intensity based on such a trend? Further on (lines 295-299), the total sand transport is calculated on the basis of a linear trend, but it is additionally cut off close to the ground where in fact the sand transport rate is the highest, provided it is not within a vegetation. I do not understand this idea. Additionally, the annual potential sand transport was calculated from Van Rijen and Strypsteen (2020) model, using the data from nearby (?) weather station and assuming a priori that it reflects given conditions and studied beachdune system well. Its outcome was verified by a comparison with the result of a single measurement of the sand transport rate (lines 300-304).

  Response: *Thank you very much for this comment. The explanation for the usage of linear interpolation of the data has been extented. We agree, that this method is not perfect, but we did not find a function, which would be applicable onto most or all measurements. As mentioned in the conclusion, we advise the usage of traps similar to the ones used in Rotnicka (2013) or Sherman et al. (2014) for measuring in between vegetation due to their seamless vertical measurement. In the description of the island and study site, we have included the positions of the metorological stations and also clarified that this ground trouthing with field data is only one single measurement.*

  For these measurements, linear interpolation is used. As most positions feature neighbouring vegetation, the transport rate directly above the surface cannot be predicted, therefore the area below the lowest trap is calculated as trapezoids. Other functions, like exponential functions or gaussians, either did not result in an adequate fit of the data points themselves or featured overfitting

- Comment 3: The measurements of sand transport rate were made during four days at relatively low wind speed (looking at Fig. 5 it was probably less than 6 m/s at 1 m height, site on the beach), so the sand transport must have been relatively weak and probably intermittent. One of the measurements was carried out throughout the night (lines 289-290), so if the BEST trap did not fill up with sand for such a long time, the transport must have occurred occasionally. Some of the measurements were accompanied by rain which probably suppressed such a weak transport to zero. The maximum sand transport rate was 300 g/m/h (Fig. 6) – it means 3 g of sand per 1 cm of beach during 1 hour. My conclusion is that the data set collected by the author does not represent sand transport conditions on the beach and cannot be used as a basis for any further calculations or model testing. And in fact they are not used further on.

  Response: *Thank you for your detailed feedback. The wind velocity was very low during during about half of the field campaign, which probably resulted in the lack of measuring sediment in this time window. We have included this in the discussion and refered to the measurements. Further, we more clearly stated, that only at the end of the first measurement slight precipitation occured and all presented data in the figure 9 was derived using the linear interpolation. The first measurement of AT1 resulted in the 5505 g $m^{-1}$ $h^{-1}$, showed in figrue 8. Only this measurement is quantitavely used for ground trouthing.*

- Comment 4: The authors compare the annual sand transport rate calculated on the basis of a model developed by Van Rijen (2018) and Van Rijen and Strypsteen (2020) with transport rate derived (how?) from the geodetic data (volume changes of beach and dune), but in Chapter 4.2 we got to know that the beach and dune was nourished with sand!! In fact, the measurement of sand transport rate presented in the manuscript are of low value, but they are unnecessary for the rest of the paper. Therefore I suggest to remove this part and better organize the article.

  Response: *Thank you for this comment. We added the explanation on the methodology of deriving annual mass changes of the dune system. Our explanation, which beach sections have been nourished in the past were superficial and have been clarified in section 3.2.. Further, we have removed the southern beach section from any further comparisons. Still, we left this beach in the figures 5, 6 and table 3 and 4, but included an explanation for the lack of further analysis in the text.*

A summation of the derived transported sediment (equation 6) delivers the total mass of annual transported sediment, which is shown in table 3. All values are shown per meter width, to enable a comparison between the two beach sections. In addition to the calculated approximation, the mass changes of the dune systems derived from the geodetic data are shown. Conversion from derived volumes to the presented mass was performed by assuming a bulk density of 1650 $kgm^{-3}$. These values are annual averages for the period from 2018 to 2022. For the northern dune system, the approximation of the total annual aeolian transported mass into the dune system exceeds the actual mass changes by about a factor of 6, whereas the western dune section only exceeds them by about 34 percent.

However, the surplus of the southern beach was not a natural development, as beach and dune nourishments were performed by the *NLWKN* to repair eroded dunes in this area. This hinders a further comparison of the before mentioned volume changes with the potential aolian transport.

*Additional comments*

- Lines 80-81 "the available fetch length between the tidal high water line and the dune foot, usually in the form of a beach"? Can it be anything other than a beach?

  Response: *Thank you for this comment. We removed the second part of the sentence.*

- Line 95 – ebda?

  Response: *Thank you for bringing this up. We removed it. We meant "at this exact position".*

- Line 97 – not to "to lateral upwind canopy coverage percentages" but to density of vegetation.

  Response: *Thank you for pointing this out. We have corrected it.*

  vegetation density

- Lines 98-99 – ranges of vegetation density – cite the source of this data.

  Response: *Thank you for pointing this out. We have added the source.*

- Line 148 – What do you mean by "primary dune row"? A foredune?

  Response: *Thank you for bringing this up. Yes we mean the foredune. We have used this term, because the fore dune (system) is sometimes used very broadly including more than the very first row of higher dunes.*

- Lines 158-159 – the first sentence is unnecessary

  Response: *Thank you for this comment. This sentence was changes during the restructuring of the manuscript.*

- Lines 171-174 – it is not clear. You used only BEST traps, didn't you?

  Response: *Thank you for this comment. This sentence also was changes during the restructuring of the manuscript.*

- Line 187 – you should provide at least an approximate grass density and height as depending on it sand transport is different

  Response: *Thank you bringing this up. We have added the height in the text.*

- Line 191 – "In total, six data sampling intervals lasting each half a day were logged". You used BEST traps – did they operate for half a day?

  Response: *Thank you for this comment. Yes, the BEST traps operated for about half a day each time. The exact times are listed in a table in the appendix.*

- Table 1 – are these values averages calculated for wind event during experiments? What was the actual wind speed at any elevation and what was its variability during the measurements?

  Response: *Thank you very much for this question. The wind velocities for the individual heights can be found in figure 2*

- Line 212 – you did the research, not Williams (1964)

  Response: *Thank you for noticing this misformulation.*

  This method has been demonstrated to be applicable for transport on the open beach (Williams, 1964), which is the case at AT1 in this study.

- Equation 4 – there is no explanation of all symbols used in the equation: QD, $\alpha$D, u*, th
  Some of them are explained further in the text, but it is too far

  Response: *Thank you pointing this out, we have added explanations*

- Line 221 – median, not medium. How did you determine the critical fetch distance?

  Response: *Thank you for this comment. We are refering to Strypsteen et al. (2024), which found a maximum critical fetch length of about 100 m*

- Line 229 – incipient velocity?

  Response: *Thank you for this question. We meant the velocity, where the critical shear velocity is reached.*

- Line 234 – replace solid with grain or particle

  Response: *Thank you for this comment. We agree and this sentence was rephrased.*

- Line 238 – what do you mean by "nearby weather station" Explain in the section on Study site.

  Response: *Thank you for this comment. We have extended our explanation of the study site including these locations.*

- Line 248 – air humidity

  Response: *Thank you for this comment. We have corrected it.*

  air humidity

- Line 284 – what is front crest?

  Response: *Thank you for this question. We mean the frontal position directly behind the frontal top edge of the dune.*

- Fig5 – use the same unit for q and Q (either grams or kilograms) here and throughout the text

  Response: *Thank you for this comment. We now use g.*

- Line 300 – do these wind data come from your measurements or from the weather station which is far from your study site?

  Response: *Thank you for pointing this out. We have extended the explanation on the data source.*

- Line 335 – explain $\alpha$dir and $\alpha$W

  Response: *Thank you for this comment. The explanation is extended.*

- Line 345-350 and Table 2 – you provided calculated sand transport rate for 3 beach sections, but they are not described in the study site. Why is Spiekeroog divided into 3 segments, what are the differences between them?

  Response: *Thank you for pointing out the missing explanation. It is added in the site description.*

- Table 2 – how did you calculate annual sand transport rate on the basis of geodetic data? You can calculate volume changes, but how did you transform it into the transport rate. Did you assume any sand bulk density? Provide it. There is not any explanations in METHODS.

  Response: *Thank you for pointing this out. An explanation is added.*

- Lines 393-395 – "For no clear reasons, the AT2 did not measure any significant transport". This site was at the base of a dune, were the airflow decelerates due to an increase in the pressure gradient (during onshore or oblique onshore winds). Analyse the data on wind direction! I can only guess that it was alongshore or oblique onshore (Figure 6). Additionally, the wind during the measurements was very weak, it's no wonder that sand was not transported there. Whether the sand transport will occur here or not depends on both the wind force and direction.

  Response: *Thank you for this comment. We have added this in the discussion. The wind was almost perfectly onshore for the northern beach section for the entire field study (left is west in the figure).*

- Conclusions – this section repeats what has been done (lines 430-440), and further on there is a kind of a discussion with literature (lines 442-459). In fact these are not conclusions. In lines 459-460 the authors stated that dunes grow that independently of their nourishment but I do not any information on the magnitude of the nourishment and therefore I do not know how the authors came to this conclusion.

  Response: *We have revised the conclusion and have removed the southern beach section, as it is the only section with nourishments.*

**References**

GDWS: DGM-W 2021 Außenweser / Unterweser: 1x1 m Raster xyz-ASCII Data set ETRS89 UTM32N: Part 1-3, Coastal Bathymetry Germany Weser, 1-3, URL `https://www.kueste ndaten.de/Tideweser/DE/Service/Kartenthemen/Kartenthemen_node.html`, 2021.

GDWS: Pegel online: Spiekeroog: 9410010, 1, URL `https://www.pegelonline.wsv.de/gas t/stammdaten?pegelnr=9410010`, 2024.

Jackson, N. L. and Nordstrom, K. F.: Aeolian sediment transport and landforms in managed coastal systems: A review, Aeolian Research, 3, 181–196, https://doi.org/10.1016/j.aeolia.2 011.03.011, 2011.

Nikuradse, J.: Strömungswiderstand in rauhen Rohren: Aus dem Kaiser Wilhelm-Institut für Strömungsforschung, Zeitrschrift für angewandte Mathematik und Mechanik, 11, 409–411, 1931.

NLWKN: Digital Elevation Model Spiekeroog: Forschungsstelle Küste, airborne topography survey, 2023, 2023.

Rotnicka, J.: Aeolian vertical mass flux profiles above dry and moist sandy beach surfaces, Geomorphology, 187, 27–37, https://doi.org/10.1016/j.geomorph.2012.12.032, 2013.

Sherman, D. J., Swann, C., and Barron, J. D.: A high-efficiency, low-cost aeolian sand trap, Aeolian Research, 13, 31–34, https://doi.org/10.1016/j.aeolia.2014.02.006, 2014.

Strypsteen, G., Delgado-Fernandez, I., Derijckere, J., and Rauwoens, P.: Fetch–driven aeolian sediment transport on a sandy beach: A new study, Earth Surface Processes and Landforms, https://doi.org/10.1002/esp.5784, 2024.

Williams, G.: Some aspects of the eolain saltation load, Sedimentology, 3, 257–287, https://doi.org/10.1111/j.1365-3091.1964.tb00642.x, 1964.